# OpenReview forum: "From Noise to Signal: Enabling Foundation-Model Pretraining on Noisy, Real-World Corpora via Quality-Aware Tokenization"
_ICLR.cc/2026/Conference — ICLR 2026 Conference Desk Rejected Submission_

### Official Review · Reviewer_nZxw · 2025-10-29

**Soundness:** 2
**Presentation:** 2
**Contribution:** 3
**Rating:** 4
**Confidence:** 3

**Summary:**

This paper introduces QA-Token, a quality-aware tokenization framework designed to incorporate domain-specific data reliability signals into the token merging process. The work frames tokenization as a bilevel optimization problem and employs reinforcement learning with Gumbel-Softmax to adaptively construct token vocabularies that are robust to noise. The method is evaluated on genomics and high-frequency financial domains, where the authors claim substantial gains over conventional tokenizers and character/byte-level baselines. The paper argues that this approach improves robustness and downstream task performance in naturally noisy domains.

**Strengths:**

- Timely and relevant motivation — modeling data quality during tokenization is a meaningful direction, especially for high-noise domains such as genomics or financial microstructure data.

- Novel perspective — explicitly incorporating quality signals into the tokenizer construction process is underexplored, and the bilevel formulation is conceptually innovative.

- Strong empirical gains reported — especially in genomics variant calling and financial forecasting; claims are accompanied by multiple downstream tasks and some statistical testing.

- No additional inference-time cost — the added complexity is isolated to tokenizer construction, which is attractive from a practical standpoint.

**Weaknesses:**

1. Baseline comparison appears incomplete and potentially unfair.
Although the paper emphasizes noise robustness as the core motivation, all experimental baselines are standard tokenizers (BPE, WordPiece, SentencePiece, etc.). There is no comparison against data denoising / filtering / quality-weighted sampling / robust preprocessing pipelines, which are widely used in both genomics and financial modeling. Since the paper positions itself as addressing noise, not purely tokenization, this omission leaves the evaluation incomplete.

2. The method does not appear truly “general” or domain-agnostic as claimed.
The framework requires explicit per-domain manual design of quality functions (e.g., Phred-based scoring for genomics vs. MI/volatility-based metrics for finance), and these formulations differ substantially. This suggests a domain-tailored architecture rather than a general-purpose quality-aware tokenizer. The current framing may be somewhat overstated relative to its actual scope.

3. Clarity and reproducibility are not yet at publishable standard.
- No code or implementation is currently released, so reproducibility cannot be assessed.
- The presentation lacks an early and clear articulation of the exact research question and evaluation setup. Benchmarks and baseline selection logic are spread across sections rather than systematically introduced.
- It is occasionally difficult to trace how the proposed tokenizer concretely integrates into pipelines like BWA-MEM → GATK in genomics, which do not typically depend on tokenization.

4. Potential risk of information leakage in the financial evaluation.
The mutual-information-based quality metric appears to condition on future returns. Unless the authors enforce strict causal rolling windows with no lookahead during tokenizer training, the reported financial results may be artificially inflated.

**Questions:**

1. Can you confirm whether q_info in the financial domain is computed using strictly causal, horizon-appropriate rolling windows without access to future returns at tokenizer construction time?

2. Why are noise-removal or robust data filtering baselines not included, given that noise handling is the central motivation of this work?

3. Would you agree that the current method is domain-tailored rather than universal? If so, would revising the framing away from a “general-purpose” claim be more accurate?

4. Could you clarify how the tokenizer interacts with the genomics variant-calling pipeline, which typically does not tokenize sequences? A schematic would be helpful.

5. Will code and tokenizer configs be released to ensure reproducibility, as strongly expected by ICLR?

---

> ### Author Response · Authors · 2025-12-04
>
> We thank the reviewer for their rigorous feedback.
>
> ## On Baseline Comparison with Denoising/Filtering Methods
>
> **QA-Token builds semantically meaningful tokens while preserving information—not merely handling noise.**
>
> | Approach | Limitation |
> |----------|------------|
> | **Denoising** | Discards signal with noise; irreversible |
> | **Filtering** | Loses 80%+ of corpus |
> | **Quality-weighted sampling** | Does not modify vocabulary structure |
> | **QA-Token** | **Down-weights noise without discarding it** |
>
> **Key distinction**: Denoising operates *before* tokenization and cannot construct tokens encoding reliability. QA-Token embeds quality directly into vocabulary via the merge score (Theorem 4, §3.3; Appendix E.2).
>
> **Data Curation Baseline** (Appendix T.2): QA-Token on noisy data (0.891 F1) outperforms BPE on clean data (0.847 F1) by **+5.2%** (p < 0.001).
>
> **Why denoising can hurt**: Low-quality regions contain meaningful signal—repetitive sequences in genomics (Appendix L.1), regime-change information in finance (Appendix L.2), intentional stylistic choices in NLP (Appendix D). See Theorem 12 (Appendix K) for information-theoretic justification.
>
> ---
>
> ## On Generality vs. Domain-Tailored Design
>
> **The core framework is fully domain-agnostic. Quality functions are modular inputs.**
>
> | Component | Reference |
> |-----------|-----------|
> | Bilevel optimization (Definition 1) | §3.2 |
> | NP-hardness (Theorem 1) | §3.2; Appendix E.5 |
> | MDP formulation (Definition 2) | Appendix H |
> | RL policy optimization | §4.1; Appendix G |
> | Gumbel-Softmax adaptation | §4.3; Appendix J |
> | Merge score (Theorem 4) | §3.3; Appendix E.2 |
> | **Quality function $q_t$** | Requires specification (§2) |
>
> All components except $q_t$ are domain-agnostic. The *mathematical framework* is general; the *quality signal* is a modular input.
>
> **Automating quality function design**: Existing standards (Phred, confidence intervals, perplexity), derived metrics (entropy, embedding coherence—Appendix F), or meta-learning from downstream gradients (future work, §7).
>
> **Revised framing** (see §2): QA-Token is a **"general framework with modular domain-specific quality inputs."**
>
> ---
>
> ## On Reproducibility
>
> Code is available at: https://anonymous.4open.science/r/qa-token
>
> See also **Appendix X.2** for our dataset release plan.
>
> ### Genomics Pipeline Integration
>
> QA-Token embeddings serve as **auxiliary features** for a classifier that re-scores GATK's candidate variants. The standard BWA-MEM → GATK pipeline runs unchanged; QA-Token adds learned representations to improve variant filtering. Details in Appendix C.1 and T.1.
>
> ---
>
> ## On Information Leakage in Finance
>
> **We guarantee strict temporal causality through a walk-forward protocol** (Appendix V.6):
>
> > *"For each segment $k$, the model was trained on data up to the start of segment $k$, validated on segment $k-1$, and tested out-of-sample only on segment $k$."* (line 1636)
>
> > *"All features, quality scores, token definitions at time $t$ based strictly on information up to $t-1$."* (line 1637)
>
> **No-MI Ablation** (Table 3): Even without MI ($R_P = 0$), QAT-QF achieves +4.4% return prediction and +12.9% Sharpe over BPE—confirming gains derive from quality-aware tokenization, not information leakage.
>
> ---
>
> ## On Questions
>
> **Q1: Is $q_{\text{info}}$ computed with strictly causal rolling windows?**
>
> Yes. The MI computation uses (Appendix V.6):
> - Trailing 30-day window that never overlaps with test data
> - NormFactor_MI is computed from training window only
> - $\tau = 5$ minutes future return horizon with 3-bin discretization
>
> **Q2: Why no noise-removal baselines?**
>
> See Appendix T.2: QA-Token on noisy data (0.891 F1) outperforms BPE on clean data (0.847 F1) by **+5.2%**.
>
> **Q3: Is the method domain-tailored rather than universal?**
>
> The mathematical framework is domain-agnostic. Only $q_t$ requires domain specification. Revised framing: **"General framework with modular domain-specific quality inputs."**
>
> **Q4: How does the tokenizer integrate with BWA-MEM → GATK?**
>
> QA-Token embeddings **augment** (not replace) the variant calling pipeline:
> - Embeddings provide features for a variant classifier
> - Classifier operates alongside GATK using its outputs as context
>
> Details in Appendix C.1 and T.1.

---

### Official Review · Reviewer_AJ7S · 2025-10-30

**Soundness:** 3
**Presentation:** 3
**Contribution:** 3
**Rating:** 8
**Confidence:** 2

**Summary:**

The paper introduces QA-Token, a quality-aware tokenizer designed for learning robust tokenization under noisy data conditions (e.g., genomics, finance, and social text. The authors formalize tokenization as a bilevel optimization problem balancing downstream model likelihood, vocabulary complexity, and a reliability reward Q(V,Z).
And the paper proposed a reinforcement learning algorithm to solve this bilevel optimization problem approximately, with convergence guarantee.
They also use Gumbel-Softmax relaxation to achieve end to end learning of quality sensitivity parameters.
Experiments on genomics, finance, and text corpora show consistent gains and reduced token length.

**Strengths:**

1. The paper targets an underexplored but practically important problem: tokenization under measurement noise, which is crucial in many domains.
2. The bilevel formulation, NP-hardness proof, and derivation of the quality-aware merge score are mathematically sound and well integrated.
3. Propose a clear reinforcement learning algorithm to implement.
4. The experiment showstrong and consistent improvements across multiple noisy domains

**Weaknesses:**

1. No significant weakness. just one minor: The approach assumes availability of per-symbol quality (e.g., Phred scores), which may not generalize to other kind noised data.

**Questions:**

1. Can you find more fields with noised data and we can easily get quality score q_t? Can we also apply this tokenizer technique to such fields?

---

> ### Author Response · Authors · 2025-12-04
>
> We thank the reviewer for their feedback and deep understanding of our work.
>
> ## On the Assumption of Per-Symbol Quality Availability
>
> **QA-Token remains valuable even without explicit quality signals.**
>
> If per-symbol quality scores are unavailable, QA-Token reduces to optimizing the **information reward** ($R_I$) and **complexity penalty** ($R_C$) components, which still construct semantically meaningful hierarchical tokens.
>
> **Ablation Evidence**: See **Table 2**.
>
> | Component Removed | F1 | Contribution |
> |-------------------|-----|--------------|
> | None (Full) | 0.891 | — |
> | $R_I = 0$ | 0.872 | +2.1% |
> | $R_Q = 0$ | 0.825 | +7.4% |
>
> **Key insight**: $R_I$ and $R_C$ contribute **independently** of quality:
>
> - Removing $R_I$ alone drops performance by 2.1% (0.891 → 0.872)
> - These components optimize for **semantically meaningful hierarchical structure** even when quality signals are removed.
>
> The improvement from $R_I$ and $R_C$ comes from:
>
> 1. **Information Reward ($R_I$)**: Maximizes mutual information between tokens and downstream targets, discovering informative merges regardless of quality.
>
> 2. **Complexity Penalty ($R_C$)**: Follows MDL principles (§3.2), penalizing $\Phi(V) = |V| \log |V| + \sum_{t \in V} |t| \cdot H(t)$.
>
> ### Theoretical Justification
>
> The information-theoretic objective (Theorem 12, Appendix K) shows that QA-Token minimizes:
>
> $$\mathcal{L}_{\text{QA}}(V) = -I(T;Y|Q) + \beta \cdot I(T;X|Q)$$
>
> When $Q$ is unavailable (uniform quality), this reduces to standard rate-distortion optimization—still superior to frequency-only BPE because it explicitly optimizes downstream relevance.
>
> **Conclusion**: Quality signals provide the largest gains (+7.4%), but $R_I$ independently contributes +2.1%. For domains without explicit quality scores, we can:
>
> 1. **Use derived quality proxies**: Entropy, embedding coherence, confidence scores
> 2. **Rely on $R_I$ and $R_C$ alone**: Still builds better structure than BPE
> 3. **Set $\alpha = 0$**: Falls back to information-optimized tokenization
>
> ---
>
> ## On Additional Fields with Noisy Data
>
> **We evaluate QA-Token on natural language processing, specifically noisy social media text.**
>
> ### Social Media Text (QA-BPE-nlp)
>
> Social media exhibits noise in the form of:
> - **Orthographic variations**: Misspellings, intentional misspellings ("gonna", "u")
> - **Character repetitions**: Emphasis ("sooooo goood")
> - **Slang and neologisms**: Platform-specific language
> - **Temporal drift**: Evolving vocabulary over time
>
> ### Quality Metrics for Social Media
>
> We define $\mathbf{q}_t^{\text{social}} = (q_{\text{orth}}, q_{\text{sem}}, q_{\text{temp}}, q_{\text{plat}})$ capturing orthographic, semantic, temporal, and platform-specific quality. Full formulation in **Appendix D**.
>
> ### TweetEval Benchmark Results
>
> Full results in **Appendix X.13**.
>
> **Key Result**: QA-BPE-nlp achieves **+2.2% absolute improvement** (70.0 vs 68.5) over SuperBPE across all tasks, demonstrating effectiveness on noisy user-generated content.

---

### Official Review · Reviewer_b3bG · 2025-10-31

**Soundness:** 2
**Presentation:** 2
**Contribution:** 2
**Rating:** 2
**Confidence:** 4

**Summary:**

This paper introduces QA-Token, a novel tokenization framework that uses reinforcement learning to construct vocabularies based on data quality. The authors verify the effectiveness of the method on multiple domains.

**Strengths:**

1. The core idea of making tokenization "quality-aware" is an important research direction.
2. The paper is very ambitious in its scope. It proposes a general framework and applies it to multiple domains.
3. The paper reports performance gains over the baselines it compares against

**Weaknesses:**

1. The proposed method is very complex and expensive compared to standard tokenizers:

(a) The authors report using 50-60 GPU-hours on a A100 for tokenization, which is just a preprocessing step. This cost is very expensive for a preprocessing step, especially compared to other tokenization methods. Moreover, the network trained for RL policy is very small. It is unclear why so much time is needed for training such a tiny network.

(b) The method uses heavy reinforcement learning and optimization techniques to solve problems that don't need it. For example: in genomics, it "rediscovers" codons as meaningful tokens. Codons have been known for a long time as the fundamental units of genetic code. You could have actually baked this information from the start into the tokenizer [1].

Ultimately, the framework feels like a solution in search of a problem. The authors have built a computationally intensive and over-engineered system that in the genomics case rediscovers established domain knowledge.

2. Ambiguity in experimental setup: the paper is not fully explicit about the training process for the foundation models. While the context implies training from scratch, this should be clearly stated.

3. While the paper does the evaluation across multiple domains, I believe each of these domains warrant a more personalized solution given their unique characteristics and complexity. It would have been better if the authors focused on a single domain, instead of pursuing the universality. For instance in genomics recently there were proposed several interesting biologically-informed architectures [2, 3].

4. Evaluation lacks depth and insight. Beyond the choice of baselines, the paper's evaluation on multiple domains prevents a deep and comprehensive evaluation within any single field. A more focused study would have allowed for a much richer and conclusive analysis whether the method is useful or not.

5. The paper's central premise is that its framework can unlock the value in noisy datasets. However, it doesn't adequately compare its approach against a good data curation baseline. It would be interesting to see an experiment where a standard tokenizer like BPE is trained only on a high-quality subset of the data.

6. Paper has **18** !!! missing references (highlighted by ?? in PDF file).

7. No code for experiments is provided in supplementary material.

[1] BioToken and BioFM – Biologically-Informed Tokenization Enables Accurate and Efficient Genomic Foundation Models, 2025 Medvedev et. al.

[2] A DNA language model based on multispecies alignment predicts the effects of genome-wide variants, 2025 Benegas et. al.

[3] A Phylogenetic Approach to Genomic Language Modeling, 2025 Albors et. al.

**Questions:**

1. Could you please confirm whether the METAGENE-1 and the 1.2B financial foundation models were trained from scratch using your new tokenizers?
2. Have you considered comparing your method in the genomics domain to a simpler knowledge-driven tokenizer that directly encodes known biological motifs like codons such as [1] referenced above?
3. The training time of 50-60 GPU-hours for the RL policy seems exceptionally high for what is described as a small network. Could you provide a breakdown of where this computational time is spent?

---

> ### Author Response · Authors · 2025-12-04
>
> We thank the reviewer for their detailed critique.
>
> ## On Computational Cost (50–60 GPU-hours)
>
> ### Detailed Cost Breakdown (Appendix X.18)
>
> | Component | Time (GPU-h) | Explanation |
> |-----------|-------------|-------------|
> | **Stage 1: RL Policy** | 30–36 | ~50 episodes × ~30K merge steps per episode |
> | **Stage 2: Adaptive Learning** | 20–24 | Gumbel-Softmax gradient through ~10 epochs |
> | **Total** | **50–60** | One-time vocabulary construction |
>
> The RL network is small (~50K params); cost lies in **environment operations**:
>
> | Bottleneck | % Time | Cause |
> |------------|--------|-------|
> | State construction | 40% | $O(|V|^2)$ pair updates per merge |
> | Episode length | 35% | ~1.6M merges (50 eps × 32K steps) |
> | Adaptive learning | 25% | Gumbel-Softmax + differentiable tokenization |
>
> **Inference**: Identical to BPE (10 ms/seq) once vocabulary is built (main text, line 466).
> **ROI**: 0.11% of METAGENE-1 training cost; >180× return via 15% token reduction (Appendix, lines 2600–2620).
>
> ---
>
> ## On "Rediscovering Codons" and Over-Engineering
>
> QA-BPE-seq received **no biological priors**, only Phred scores. The emergence of codon-aligned tokens (Appendix O) without explicit supervision demonstrates that quality signals enable recovery of meaningful structure:
>
> > *"Analysis of generated vocabularies reveals that QA-BPE-seq creates tokens aligned with biological units (codons, motifs) while breaking at error-prone junctions—a behavior that emerges without explicit biological supervision."* (line 1139)
>
> ### Why Not Hard-Code Domain Knowledge?
>
> If we hard-coded codons:
> - We could not adapt to non-coding regions (introns, UTRs, regulatory elements)
> - We could not discover novel motifs beyond current biological knowledge
> - We could not apply the same framework to finance or NLP
>
> ### Comparison to BioToken
>
> We compare against BioToken (Medvedev et al., 2025), which hard-codes biological motifs
>
> | Method | Approach | Variant F1 | Domain Scope |
> |--------|----------|------------|--------------|
> | BioToken | Hard-coded codons/motifs | ~0.86* | Genomics only |
> | DNABERT-k | K-mer tokenization | 0.851 ± 0.003 | Genomics only |
> | **QA-BPE-seq** | Quality-aware learning | **0.891 ± 0.004** | Any domain |
>
> **Key distinction**: BioToken requires domain expertise; QA-Token learns structure from quality signals alone.
>
> ---
>
> ## On Foundation Model Training: From Scratch
>
> **Yes, all foundation models are trained from scratch.**
>
> From the main text (§6.1):
> > *"Re-tokenized METAGENE-1 (7B parameters, 1.7T base pairs) with identical architecture/hyperparameters, comparing BPE vs QA-BPE-seq."* (line 344)
>
> | Model | Parameters | Tokenizer (vocab size) | Reference |
> |-------|------------|------------------------|-----------|
> | **METAGENE-1** | 7B | QA-BPE-seq (32K tokens) | §6.1 |
> | **Financial FM** | 1.2B | QAT-QF (16K tokens) | §6.2 |
> | **NLP Models** | 350M | QA-BPE-nlp (32K tokens) | Appendix D |
>
> All models use identical architectures and hyperparameters within each comparison; **only tokenizers differ**.
>
> ---
>
> ## On Domain Specialization vs. Universality
>
> One merge score (Theorem 4, §3.3, lines 188–190) applies across all domains; only $q_t$ changes (Genomics: Phred; Finance: stability/liquidity; NLP: orthographic/semantic).
>
> | Domain | Metric | Improvement | Reference |
> |--------|--------|-------------|-----------|
> | Genomics | Variant F1 | +8.1% | Table 1 |
> | Finance | Sharpe | +30.3% | Table 4 |
> | NLP | TweetEval | +2.2% | Appendix X.13 |
>
> ### Complementary to Domain-Specific Methods
>
> QA-Token is **orthogonal** to domain-specific architectures (BioToken, Benegas et al., Albors et al.). One can combine QA-Token's quality-aware tokenization with biologically-informed downstream models for potentially even greater benefits.
>
> ---
>
> ## On Evaluation Depth
>
> **We provide domain-specific depth in the appendix; the main text demonstrates breadth.**
>
> ### Appendix: Depth per Domain
>
> | Domain | Appendix | Key Content |
> |--------|----------|-------------|
> | Genomics | F, O, T.3 | Quality metrics, vocabulary analysis, real-world datasets |
> | Finance | T.4, T.5, V.6 | Multi-asset (AAPL), rolling-window, walk-forward protocol |
> | NLP | D, S, X.13 | Social media quality metrics, TweetEval results |
>
> ---
>
> ## On Data Curation Baseline
>
> We train standard BPE on only the top 20% highest-quality genomic sequences (Phred ≥ 30) and compare against QA-Token on the full noisy corpus for variant calling.
>
> **Key Result** (Appendix T.2): QA-Token on noisy data achieves **+5.2% higher variant calling F1** than BPE on curated data (0.891 vs 0.847, p < 0.001).
>
> ---
>
> ## On Missing References and Code Availability
>
> We apologize! All missing references and cross-references are now fixed in the revised PDF.
>
> **Code is available at:** https://anonymous.4open.science/r/qa-token

---

### Official Review · Reviewer_G2mF · 2025-11-01

**Soundness:** 3
**Presentation:** 3
**Contribution:** 3
**Rating:** 6
**Confidence:** 4

**Summary:**

This paper argues that tokenization should account for data quality when corpora are noisy, and proposes QA-Token, a framework that builds vocabularies with explicit quality signals. The method formalizes tokenization as a bilevel objective balancing downstream LM performance, vocabulary complexity, and reliability, derives a quality-aware merge score, uses an RL policy to pick merges under a multi-objective reward, and then learns quality-sensitivity parameters with a Gumbel-Softmax relaxation. Domain instantiations for genomics and quantitative finance show sizable gains. On simulated and real genomics tasks, QA-BPE-seq improves variant calling and related benchmarks, in finance, QAT-QF improves predictive and trading metrics; and at foundation scale, re-tokenizing a 7B metagenomic model lifts pathogen detection while reducing tokens.

The claim is that once the vocabulary is built, there is no extra inference-time cost, so the one-off training overhead amortizes over large deployments.

**Strengths:**

The paper presents a coherent end-to-end story linking a well-motivated problem to concrete algorithmic choices and convincing empirical payoffs. Theoretical framing (bilevel objective, NP-hardness, approximation via RL, and two-timescale adaptive learning) provides a principled backbone and clarifies why frequency-only tokenization is brittle under heterogeneous noise.

The derived merge score usefully factors association strength with a concave quality aggregator and domain constraints, aligning the math with the intuition that high-quality regions should be merged more aggressively.

Empirically, the genomics section is notably strong. Improvements are consistent across diverse tasks and the foundation-scale experiment demonstrates that retokenization can both reduce the token budget and improve accuracy.

The finance instantiation is also thoughtfully engineered, with a clear slate of baselines and a broad metric suite, and the paper is upfront about compute and runtime characteristics, including that inference speed matches standard BPE once the vocab is fixed.

**Weaknesses:**

The finance setup risks information leakage because one of the quality components directly uses mutual information with future returns, unless those statistics are computed strictly within the training window and never refreshed with validation/test periods, the tokenizer could be indirectly exposed to test labels. The paper should make this protocol airtight and quantify the impact of removing this component. Beyond leakage, the finance validation is concentrated on BTC/USD for a single quarter; results may be regime- or asset-specific, and the broader coverage hinted at in appendices would be more convincing in the main paper. While the theory is well-developed, many assumptions (stability, boundedness, adaptive submodularity conditions) sit in the appendices; surfacing the practical scope and likely failure modes would help readers understand when guarantees apply. Finally, sensitivity and cost–benefit analyses feel underplayed given the extra training overhead. Readers would benefit from explicit curves showing performance vs. quality-sensitivity parameters, batch size, and vocabulary size, and from a clearer return-on-investment picture that relates RL/adaptive hours to downstream gains at different scales.

**Questions:**

- How do you guarantee no look-ahead when computing the “information quality” term in finance, which depends on mutual information with future returns? A strict train-only estimation and a “no-MI” ablation would de-risk leakage and clarify how much this component drives gains.

- Can you promote the cross-asset and out-of-sample extensions now in the appendix (e.g., AAPL) into the main text and add rolling-window results? This would demonstrate robustness beyond a single asset/regime.

- Could you provide sensitivity curves for the learned quality exponent and other adaptive parameters, as well as batch size and vocabulary size? This would help practitioners tune QA-Token and understand stability.

- Which core assumptions behind the approximation and convergence guarantees are most likely to be violated in practice, and how should users detect and mitigate those cases? A brief “when guarantees fail” section in the main text would increase trust.

---

> ### Author Response · Authors · 2025-12-04
>
> We thank the reviewer for the thorough and constructive feedback.
>
> ## On Information Leakage Risk in Finance ($q_{\text{info}}$ with MI)
>
> **We guarantee strict temporal causality through a rigorous walk-forward protocol.**
>
> We address the concern about information leakage comprehensively in **Appendix V.6**, which explicitly documents our backtesting protocol:
>
> ### Protocol Details (Appendix V.6):
>
> 1. **Walk-Forward Validation** (line 1636): *"For each segment $k$, the model (including the QA-Token vocabulary construction and downstream predictive/trading model) was trained on data up to the start of segment $k$, validated on segment $k-1$, and then tested out-of-sample only on segment $k$."*
>
> 2. **Lookahead Bias Prevention** (line 1637): *"All features, quality scores, token definitions, and trading decisions at any time $t$ were based strictly on information available up to and including time $t-1$."*
>
> 3. **MI Normalization**: The `NormFactor_MI` is computed exclusively from the **training window**—never refreshed with validation/test data.
>
> 4. **Test Set Isolation** (line 1638): *"The final 15% (~2 weeks) was held out and used only once after all model development and tuning."*
>
> ### No-MI Ablation
>
> The $R_P$ reward component is the **only** place MI appears in QAT-QF. We ablate this completely by setting $R_P = 0$.
>
> **Results**: See **Table 3**. Even without MI ($R_P = 0$), QAT-QF outperforms BPE by +4.4% return prediction and +12.9% Sharpe—demonstrating gains derive from quality-aware tokenization, not MI.
>
> ---
>
> ## On Single Asset/Quarter Limitation
>
> **Cross-asset and extended temporal results exist in the appendix.**
>
> ### Cross-Asset: AAPL High-Frequency Equities
>
> See **Appendix T.4** for complete AAPL results. Key finding: QAT-QF achieves **+28.4% Sharpe improvement** over BPE (1.81 vs 1.41) on equities—consistent with crypto results.
>
> ### Rolling-Window Out-of-Sample (BTC/USD, Full Year 2023)
>
> See **Appendix T.5** for quarterly breakdown. Key findings:
> - **Q1–Q4 2023 average**: QAT-QF Sharpe 1.61 vs BPE 1.32 (**+29.8%**)
> - **Q3 2023** (high volatility): Gains persist (+26.1%), demonstrating **cross-regime robustness**
> - Consistent improvements (+26–31%) across all market regimes
>
> ---
>
> ## On Assumptions and Failure Modes
>
> ### Core Assumptions
>
> See **Appendix E.6** for complete formal assumptions (A1–A4):
> - **A1** (Bounded Frequencies): Always satisfied for finite corpora
> - **A2** (Bounded Qualities): By construction (sigmoid/clipping)
> - **A3** (Bounded Rewards): EMA normalization enforces boundedness
> - **A4** (Regular Learning Rates): Standard $\eta_t = O(1/\sqrt{t})$ schedule
>
> ### Robustness Guarantee
>
> See **Theorem 17** (Appendix X.17) for the robustness bound:
> $$\mathcal{L}(\tilde{q}) - \mathcal{L}(q) \leq \alpha \cdot \sigma_\xi \cdot \sqrt{\mathbb{E}[\|\nabla_q \mathcal{L}\|^2]}$$
>
> **Empirical Validation** (Appendix X.17):
>
> | Condition | Effect |
> |-----------|--------|
> | 20% quality noise | −4.2% genomics, −5.8% finance |
> | Adversarial quality | Matches BPE (safe fallback) |
> | 50% missing quality | Frequency-only fallback |
>
> ---
>
> ## On Sensitivity and Cost-Benefit Analyses
>
> ### Quality Sensitivity $\alpha$
>
> See **Appendix M** (main) and **Appendix X.7** (extended) for full hyperparameter sensitivity analysis. Key findings:
> - **Genomics**: Optimal $\alpha^* = 1.37$, robust across $[0.5, 2.0]$
> - **Finance**: Optimal $\alpha^* = 0.95$, robust across $[0.25, 1.50]$
>
> ### Vocabulary Size Sensitivity
>
> See **Appendix X.7** for vocabulary size ablation. Optimal trade-off at 32K vocabulary (0.891 F1, 115 tokens/seq, 55 GPU-h).
>
> ### Cost-Benefit Analysis
>
> See **Appendix X.18** for full details.
>
> | Metric | BPE | QA-Token | Δ |
> |--------|-----|----------|---|
> | Training | 5–10 min (CPU) | 50–60 GPU-h | — |
> | Inference | 10 ms/seq | 10 ms/seq | **0%** |
> | Variant F1 | 0.824 | 0.891 | **+8.1%** |
> | Sharpe | 1.32 | 1.72 | **+30.3%** |
>
> **ROI for METAGENE-1**: QA-Token = 0.11% of total training cost; 15% token reduction → **>180× ROI**.

---

### Author Response · Authors · 2025-12-04

We thank all reviewers for their thorough evaluation and constructive feedback. We address every concern through substantial new experiments and clarifications. Below, we summarize the key contributions of our rebuttal.

---

## Summary of New Evidence

1. **No-MI Ablation (Information Leakage):** The No-MI ablation (**Table 3**) sets $R_P = 0$, completely removing the mutual information component. Result: QAT-QF still outperforms BPE by **+4.4% return prediction** and **+12.9% Sharpe**—proving gains derive from quality-aware tokenization, not lookahead bias.

2. **Cross-Asset Generalization:** We evaluate on **AAPL high-frequency equities** (**Table 19**), achieving **+28.4% Sharpe improvement** (1.81 vs 1.41)—demonstrating gains transfer beyond the original BTC/USD asset.

3. **Temporal Robustness:** Rolling-window evaluation across **Q1–Q4 2023** (**Table 20**) shows **+29.8% average Sharpe** across all market regimes, with consistent gains during high-volatility Q3 (+26.1%).

4. **Data Curation Baseline:** We compare BPE trained on the top 20% highest-quality sequences (Phred ≥ 30) against QA-Token on the full noisy corpus. Result: QA-Token on noisy data achieves **+5.2% higher F1** than BPE on curated clean data (0.891 vs 0.847, p < 0.001; **Table 16**).

5. **Formal Assumptions:** We explicitly document assumptions **A1–A4** (Appendix E.6) with precise mathematical conditions for all theoretical guarantees.

6. **Robustness Guarantees:** **Theorem 17** (Appendix X.17) bounds degradation under quality corruption. Empirical validation: −4.2% under 20% noise (graceful), safe fallback to BPE-equivalent under adversarial conditions.

## New Experimental Results

| Category | New Experiment | Key Result | Reference |
|----------|----------------|------------|-----------|
| **Finance** | No-MI ablation ($R_P = 0$) | +4.4% return pred., +12.9% Sharpe over BPE | Table 3 |
| **Finance** | AAPL high-frequency equities | +28.4% Sharpe (1.81 vs 1.41) | Table 19 |
| **Finance** | Rolling-window Q1–Q4 2023 | +29.8% Sharpe across all regimes | Table 20 |
| **Genomics** | Data curation baseline | +5.2% F1 (QA-Token on noisy vs BPE on clean) | Table 16 |
| **Robustness** | Quality noise injection (20%) | −4.2% genomics, −5.8% finance (graceful) | Appendix X.17 |
| **Robustness** | Adversarial quality inversion | Safe fallback to BPE-equivalent | Appendix X.17 |
| **Theoretical** | Formal assumptions A1–A4 | Explicit conditions documented | Appendix E.6 |
| **Theoretical** | Robustness bound (Theorem 17) | Bounded degradation under quality corruption | Appendix X.17 |

---

## Quick Reference Guide

| Concern | Resolution | Reference |
|---------|------------|-----------|
| Information leakage in finance | Walk-forward protocol + No-MI ablation | Appendix V.6; Table 3 |
| Single asset/quarter limitation | AAPL + rolling-window Q1–Q4 2023 | Tables 19, 20 |
| Assumptions and failure modes | Formal A1–A4 + Theorem 17 + empirical validation | Appendix E.6, X.17 |
| Computational cost justification | Detailed breakdown + ROI analysis | Appendix X.18 |
| Domain generality | Three domains + unified merge score (Theorem 4) | Tables 1, 4; Appendix X.13 |
| Data curation baseline | BPE on clean vs. QA-Token on noisy | Table 16 |
| Reproducibility | Code + dataset release plan | Appendix X.2 |

---

## Formatting and Reproducibility
- **Dataset release plan** detailed in Appendix X.2, including tokenizer artifacts, evaluation suites, and Docker containers.
- **Our Reproducibility statement** in the main text specifies all hyperparameters, statistical methodology (10 trials, 95% CIs, Holm-Bonferroni correction), and complete proofs for all theorems.

---

We again thank all reviewers for their feedback.

---

### Note · Program_Chairs · 2026-01-17
**Submission Desk Rejected by Program Chairs**

The following references in this submission do not refer to real documents and/or have major errors in bibliographic information:

 Carl Allen Meyer and Mrinmaya Sachan. Joint learning of sentence segmentation and representation. In Findings of the Association for Computational Linguistics: EMNLP 2023, pp. 12315-12330, 2023.

Guillaume Lample, Ludovic Denoyer, and Marc'Aurelio Ranzato. Fast hierarchical language modeling. In International Conference on Learning Representations, 2018.

Dimitri P Bertsekas. Reinforcement learning: An introduction. MIT Press, 2019.

Jane Doe and John Smith. Gentokenizer: A specialized tokenizer for genomic sequences, 2023.

Zeyu Ding, Baolin Wang, Xiaoyu Wang, Guangwu Hu, Kai Chen, and Qi Chen. Towards understanding the robustness of large language models against spelling errors. In Findings of the Association for Computational Linguistics: EMNLP 2023, pp. 7891-7904, 2023.

Jacob Eisenstein. Bad characters: Imperfect ocr scanning and the hidden perils of character-level models for sequence labeling. In Proceedings of the 2013 Conference on Empirical Methods in Natural Language Processing, pp. 1734-1744, 2013.

Ming Yu et al. Direct advantage policy optimization. arXiv preprint, 2025.

BPE Super and Multiple Authors. Superbpe: Superposition prompting for autoregressive byte-level models. arXiv preprint arXiv:2401.00000, 2024.

Lei Zheng, Xiang Zheng, and Zhong Wang. Adaptive input representations for neural language modeling. In Proceedings of the AAAI Conference on Artificial Intelligence, volume 38, pp. 21163-21171, 2024.